# 1,2-$^{13}$C$_2$-Glucose Tracing Approach to Assess Metabolic Alterations of Human Monocytes under Neuroinflammatory Conditions

Ginevra Giacomello [1,*] , Carolin Otto [2], Josef Priller [3,4,5,6], Klemens Ruprecht [2], Chotima Böttcher [3,7,8,†] and Maria Kristina Parr [1,*,†] 

1   Institute of Pharmacy, Freie Universität Berlin, Königin-Luise-Str. 2 + 4, 14195 Berlin, Germany
2   Department of Neurology, Charité—Universitätsmedizin Berlin, Corporate Member of Freie Universität Berlin and Humboldt-Universität zu Berlin, 10117 Berlin, Germany
3   Department of Neuropsychiatry and Laboratory of Molecular Psychiatry, Charité—Universitätsmedizin Berlin, Corporate Member of Freie Universität Berlin and Humboldt-Universität zu Berlin, 10117 Berlin, Germany
4   German Center for Neurodegenerative Diseases (DZNE), 10117 Berlin, Germany
5   Department of Psychiatry and Psychotherapy, School of Medicine, Technical University Munich, 81675 Munich, Germany
6   UK Dementia Research Institute (UK DRI), University of Edinburgh, Edinburgh EH16 4SB, UK
7   Experimental and Clinical Research Center, a Cooperation between the Max Delbrück Center for Molecular Medicine in the Helmholtz Association and Charité—Universitätsmedizin Berlin, 13125 Berlin, Germany
8   Max Delbrück Center for Molecular Medicine in the Helmholtz Association (MDC), 13125 Berlin, Germany
*   Correspondence: ginevra@zedat.fu-berlin.de (G.G.); maria.parr@fu-berlin.de (M.K.P.); Tel.: +49-30-838-57686 (M.K.P.)
†   These authors contributed equally to this work.

**Abstract:** Neuroinflammation is one of the common features in most neurological diseases including multiple sclerosis (MScl) and neurodegenerative diseases such as Alzheimer's disease (AD). It is associated with local brain inflammation, microglial activation, and infiltration of peripheral immune cells into cerebrospinal fluid (CSF) and the central nervous system (CNS). It has been shown that the diversity of phenotypic changes in monocytes in CSF relates to neuroinflammation. It remains to be investigated whether these phenotypic changes are associated with functional or metabolic alteration, which may give a hint to their function or changes in cell states, e.g., cell activation. In this article, we investigate whether major metabolic pathways of blood monocytes alter after exposure to CSF of healthy individuals or patients with AD or MScl. Our findings show a significant alteration of the metabolism of monocytes treated with CSF from patients and healthy donors, including higher production of citric acid and glutamine, suggesting a more active glycolysis and tricarboxylic acid (TCA) cycle and reduced production of glycine and serine. These alterations suggest metabolic reprogramming of monocytes, possibly related to the change of compartment (from blood to CSF) and/or disease-related. Moreover, the levels of serine differ between AD and MScl, suggesting different phenotypic alterations between diseases.

**Keywords:** monocytes; metabolism; cerebrospinal fluid; glycolysis; tricarboxylic acid cycle; multiple sclerosis; Alzheimer; metabolites; neuroinflammation

## 1. Introduction

Neurodegenerative and inflammatory diseases such as Alzheimer's disease (AD) and multiple sclerosis (MScl) are continuously gaining focus from the scientific and public audience due to the high incidence and devastating consequences in terms of life quality and care-need of patients. Although the course, symptoms, causes, and characteristics of these neurological diseases are highly heterogeneous, some aspects are in common, i.e., the presence of neuroinflammatory features. Both diseases are characterized by microglial

activation [1–4], increased production of pro- and anti-inflammatory cytokines [5–8], and infiltration of peripheral immune cells from the bloodstream into the cerebrospinal fluid (CSF) due to the impairment of the blood–CSF barrier [9–12]. However, it remains largely unknown to what extent changing the compartment environment (e.g., from blood to CSF) influences the phenotypes and functions of this myeloid cell population. Previous studies demonstrated the phenotypic diversity of CSF infiltrating monocytes [13–17], but little is known about related changes in functions and/or changes in the metabolism of these cells compared to their counterparts in the peripheral blood. Changes in metabolism, e.g., glucose metabolism, can give a hint on cell activation and/or responses to pathology [18–20].

One of the hypotheses is that once the monocytes enter the CSF through the choroid plexus (CP), they undergo phenotypic changes and take part in the inflammatory response to eventually help the resolution of the innate inflammation [21,22]. The first reaction is to contribute to neuroinflammation to regulate adaptive immunity. Why, in chronic neuroinflammation such as AD or MScl, the resolution of the inflammation does not happen and causes neuronal damage needs to be further investigated. Similarly, whether this is caused by the aggravated impairment of the barrier between CSF and blood, by the disease, or is caused by both, needs additional investigation. Varvel et al. proposed as beneficial the inhibition of the entrance of monocytes into the brain to mitigate the inflammation occurring after seizures in status epilepticus [23], but since the function of myeloid cells, in normal conditions, is also to remove waste products (β-amyloid included), a pathologic environment may also play a role in it. Many studies have focused lately on recruited myeloid cells to better understand their role in acute or chronic neuroinflammation, and to evaluate the changes that they undergo once they change the compartment.

To have a more complete overview of the changes that CSF-infiltrating monocytes undergo, it is necessary to also consider immunometabolism and its alterations. Lately, it has been pointed out as fundamental in the regulation between pro- and anti-inflammatory profiles [24]. The metabolism of cells is reported to regulate energetic production, intracellular signaling, the well-being of the cells, and the synthesis of amino acids and other metabolites. Furthermore, glycolysis is an essential metabolic pathway for the differentiation of macrophages [25] and the activation of immune cells during inflammation [24]. To study phenotypic and metabolic changes in circulating immune cells under different conditions including experimental settings, several state-of-the-art technologies can be used such as mass cytometry (CyTOF) [26–29], cell bioenergetic analysis (Seahorse) [30–33], multiplex immunoassay (Luminex) [28,34,35], nuclear magnetic resonance (NMR) [36–38], high-performance liquid chromatography (HPLC), or gas chromatography (GC) hyphenated with mass spectrometry (MS) or tandem mass spectrometry (MS/MS) [39–42]. In-depth characterization of the metabolic pathway is achieved using HPLC- or GC-MS(/MS). The use of stable isotopic tracing methods (e.g., $^{13}$C-glucose tracing) increases confidence in identifying the metabolites affected by the study conditions or experimental manipulations.

In our previous study [21], we characterized the phenotypic alterations of infiltrating myeloid cells in the CSF of healthy individuals and patients with neurological disorders such as AD, mild cognitive impairment (MCI), and Huntington's disease. Our findings suggested activation and inflammatory response of myeloid cells as well as metabolic changes after exposure to CSF, with a slightly increased phenotypic alteration in the case of AD-CSF [21]. Using Seahorse assays, we evaluated the extracellular acidification rate (ECAR), a value linked to glycolysis, and the oxygen consumption rate (OCR), connected to cell respiration, of myeloid cells in different conditions: without stimulation or with the stimulation of CSF of healthy donors or patients with AD or MCI. The results showed an increased ECAR when the cells were incubated in the presence of CSF [21,43]. It is tempting to speculate that, with higher intensity of neuroinflammatory conditions such as in MScl, strong changes in phenotypes and metabolism will be detectable, which can possibly be a drawback to functional changes in these cells in CSF.

We also showed, using the previously validated 1,2-$^{13}$C$_2$-glucose tracing experiment [44], how the presence of CSF of AD patients in the incubation of monocytes could cause

a decrease in the conversion of [13]C pyruvate to [13]C lactate and a reduced production of serine [21].

In this study, we aim to comparably evaluate the overall metabolism of monocytes in CON and AD individuals in comparison to MScl with more pronounced neuroinflammation. We performed a 1,2-[13]C$_2$ glucose tracing experiment on monocytes targeting key-role metabolites of the major metabolic pathways of the cell (glycolysis, pentose-5-phosphate pathway, serine and glycine production, tricarboxylic acid (TCA) cycle, lactate production from pyruvate, and glutamine/glutamic acid metabolism). Intracellular and secreted unlabeled and [13]C glucose-derived metabolites were identified and quantified using HPLC-MS/MS. In this study, we identified the rewiring of the glucose metabolism of monocytes after CSF treatment, and the differential metabolic alterations in neuroinflammatory conditions such as MScl (MScl-CSF treatment) in comparison with neurodegeneration conditions such as AD (AD-CSF treatment).

## 2. Materials and Methods

This study was registered and approved by the Ethics Commission of Charité—Universitätsmedizin Berlin (Ethikkommission der Charité—Universitätsmedizin Berlin; registration number EA1/187/17), Berlin, Germany.

### 2.1. Primary Human Monocyte Isolation and Incubation

PBMCs were obtained from the German Red Cross and stored at $-80$ °C until incubation. Monocytes were then isolated from PBMCs with MACS (Pan Monocyte Isolation Kit (human), Miltenyi Biotec, Bergisch Gladbach, Germany).

After isolation, monocytes were incubated in a 24-well/plate in four different conditions with three biological replicates for each group: medium only without stimulation (3 × NS), with the addition of 30% in volume of CSF of healthy individuals (3 × CON), with the addition of 30% in volume of CSF from AD patients (3 × AD), and with the addition of 30% in volume of CSF of MScl patients (3 × MScl). Before use, pH values of all CSF samples were measured. Only CSF with a pH of about 7 was used in this study.

The monocytes were incubated for 5 h at 37 °C, with 5% $CO_2$, in a medium without glucose, pyruvate, glutamine, and phenol red (DMEM, Thermo Fisher Scientific, Inc., Waltham, MA, USA) added with labeled glucose (D-1,2-[13]C$_2$ glucose, Sigma Aldrich, Taufkirchen, Germany) to a final concentration of 4.5 g/L, and with 10% FBS (heat-inactivated, Gibco™, NY, USA). As the incubation of monocytes in a medium without pyruvate, glutamine, and phenol red may be challenging, the well-being of the cells needs to be monitored with a microscope at regular time intervals.

Details of isolation and culture of monocytes are described elsewhere [44].

### 2.2. Metabolite Extraction and Sample Preparation
#### 2.2.1. Culture Medium

After the incubation, the cell suspension was transferred to Eppendorf tubes and centrifuged for 10 min at 4 °C at 300× $g$. Then, the supernatant (cell-free culture medium, representing the secreted level of metabolites) was separated from the cell pellet and centrifuged at 15,000× $g$, for 10 min at 4 °C before the analysis with HPLC-MS/MS.

#### 2.2.2. Cell Extract

After shock-freezing of the cell pellets using liquid nitrogen 100 µL of $H_2O$: acetonitrile (ACN) (1:1, *v:v*) were added for the extraction. After thorough vortexing and incubation on ice for 5 min, the samples were centrifuged at 15,000× $g$ for 10 min at 4 °C. Aliquots of 75 µL of the supernatant without disturbing the cell remainder were used to obtain the cell extract (representing an intracellular level of metabolites). Samples were stored at $-80$ °C until the HPLC-MS/MS analysis.

Details of monocyte extraction and sample preparation are previously described in detail [44].

## 2.3. HPLC-MS/MS Setup and Analysis

The chromatographic conditions, the MS/MS parameters, and the dynamic multiple reaction monitoring (dMRM) method were optimized and validated. Details of the analytical method are reported previously [44].

Standard solutions for calibration levels and quality controls (QCs) were prepared in serial dilution from stock solution (1 mg/mL) in Milli-Q $H_2O$:ACN (1:1).

Analyses of samples, QC, and calibration levels were conducted with an Agilent 1290 Infinity II HPLC system coupled with an Agilent 6495 QqQ mass spectrometer (MS/MS) with an Agilent jet stream source with electrospray ionization (AJS-ESI), both controlled by MassHunter 10 Data Acquisition software (Agilent Technologies, Waldbronn, Germany). For the separation of the metabolites, an Agilent InfinityLab Poroshell 120 HILIC-Z column (PEEK-lined, 2.1 mm × 100 mm, 2.7 μm) was used. To optimize the chromatographic peak shape of citric acid and phosphates, passivation of the system was needed with an overnight wash of the system with 0.5% $H_3PO_4$ in ACN:$H_2O$ (9:1). Mobile phase A was 10 mM $CH_3COONH_4$ in $H_2O$ with the addition of InfinityLab deactivator additive (Agilent Technologies, Waldbronn, Germany). Mobile phase B was 10 mM $CH_3COONH_4$ in ACN with the addition of InfinityLab deactivator additive (Agilent Technologies). We added 1 mL of additive for 1 L of mobile phase. HPLC separation was achieved by running a linear gradient from 10% to 40% to 10% mobile phase A in 20 min. The optimized dynamic multiple reaction monitoring (dMRM) was conducted by applying both positive and negative ionization mode.

For further specifications on HPLC conditions, MS parameters, and mobile phase preparation, refer to Giacomello et al. [44].

## 2.4. Data Analysis

Chromatographic and spectrometric results were evaluated with MassHunter 10 Quantitative Analysis program G3336 (Agilent Technologies, Waldbronn, Germany). For graphical summaries, Prism 7 (GraphPad) was used.

For the statistical analysis, OriginPro 2021b (OriginLab) was used. Normal distribution and equality of variances were evaluated with Shapiro–Wilk test and Levene's test, respectively. For the evaluation of the significant variation between groups, a one-way ANOVA was conducted for normally distributed groups and Kruskal–Wallis for those not normally distributed. To determine which group was significantly different, a Tukey's test or a Dunn's test (non-parametric) was conducted.

## 3. Results

Using our validated method [44], unlabeled pyruvate, lactate, glycine, glutamine, serine, glutamic acid, and citric acid, as well as 2,3-$^{13}C_2$ pyruvate, 1,2-$^{13}C_2$ lactate, 2-$^{13}C$ glycine, and 1,2-$^{13}C_2$ glutamine produced by human primary monocytes, incubated in vitro without stimulation or with the addition of CSF of healthy donors or patients with AD or MScl, were successfully quantified. Of note, the concentrations of 2,3-$^{13}C_2$ serine and 1,2-$^{13}C_2$ citric acid were lower than the limit of quantitation (LOQ), and thus they were considered detectable but not quantified metabolites. This confirmed the integration and conversion of the supplemented 1,2-$^{13}C_2$ glucose into the glucose metabolism of the primary monocytes. ATP, AMP, acetyl coenzyme A (AcCoA), fructose-6-phosphate, glyceraldehyde-3-phosphate, phosphoglyceric acid, and ribose-5-phosphate concentrations were below the limit of detection (LOD) under our in vitro experimental conditions. These results are summarized in Figure 1.

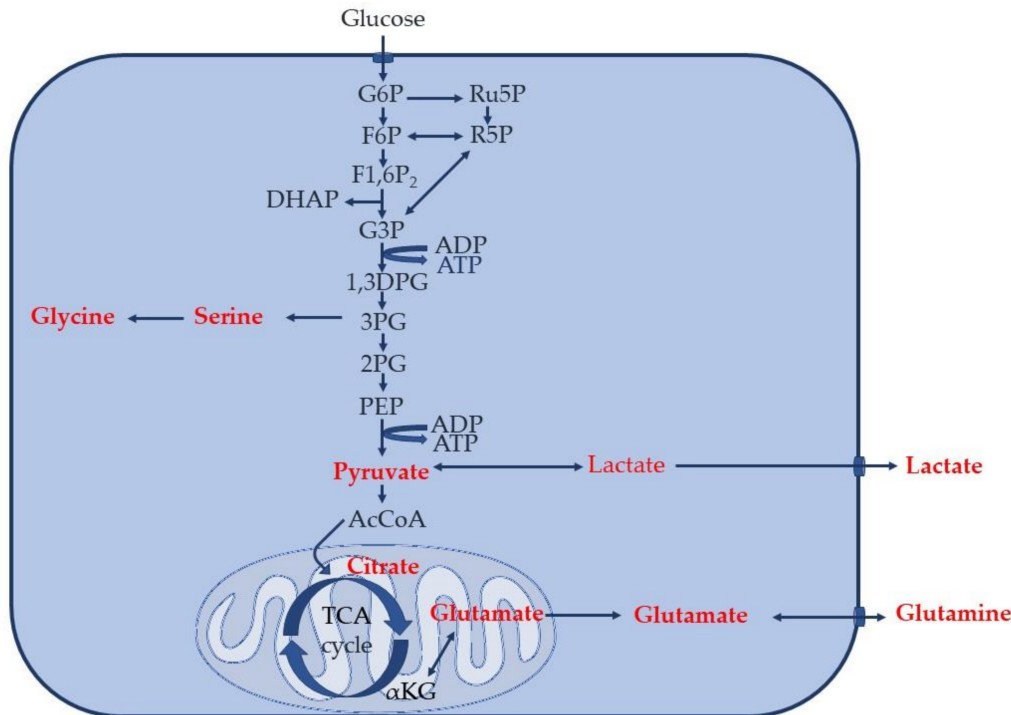

**Figure 1.** A common glucose metabolism in the cells. Metabolites that are detectable and quantifiable in this study are labeled in red (i.e., whose concentrations are at or higher than LOD and LOQ).

As common in HILIC separation [45–47] and as already reported previously [44], it is not possible to chromatographically separate and, therefore, unambiguously distinguish between citric and isocitric acid under our measurement conditions. Thus, hereafter citric acid refers to a mixture of the two isomers. The individual concentrations of all the analytes (labeled and unlabeled) that have been quantified in cell lysate and medium are summarized in Supplementary Table S1.

### 3.1. Increased Glucose Conversion in Monocytes after Exposure to CSF

As shown in Figure 2, increased intracellular concentrations of citric acid ($p$ = 0.0073), glutamine ($p$ = 0.0004), and pyruvate ($p$ = 0.0387) were detected in monocytes after treatment with CSF (i.e., CON-, AD-, and MScl-CSF) in comparison to non-stimulated (NS) monocytes. In contrast, serine ($p$ = 0.0356) and glycine ($p$ = 0.0166) were quantified at a lower concentration compared to the NS group. These observations were partly confirmed with post hoc tests (Tukey's test for normally distributed and Dunn's test for non-normally distributed data), i.e., the concentrations of intracellular citric acid and glutamine were significantly increased in monocytes after exposure to CON-CSF, AD-CSF, and MScl-CSF in comparison with the NS group, whereas pyruvate concentration was significantly increased only in the CON-CSF-treated group compared with NS. Intracellular serine and glycine concentrations were finally significantly different only between AD-CSF-treated and MScl-CSF-treated monocytes, each compared to the NS group.

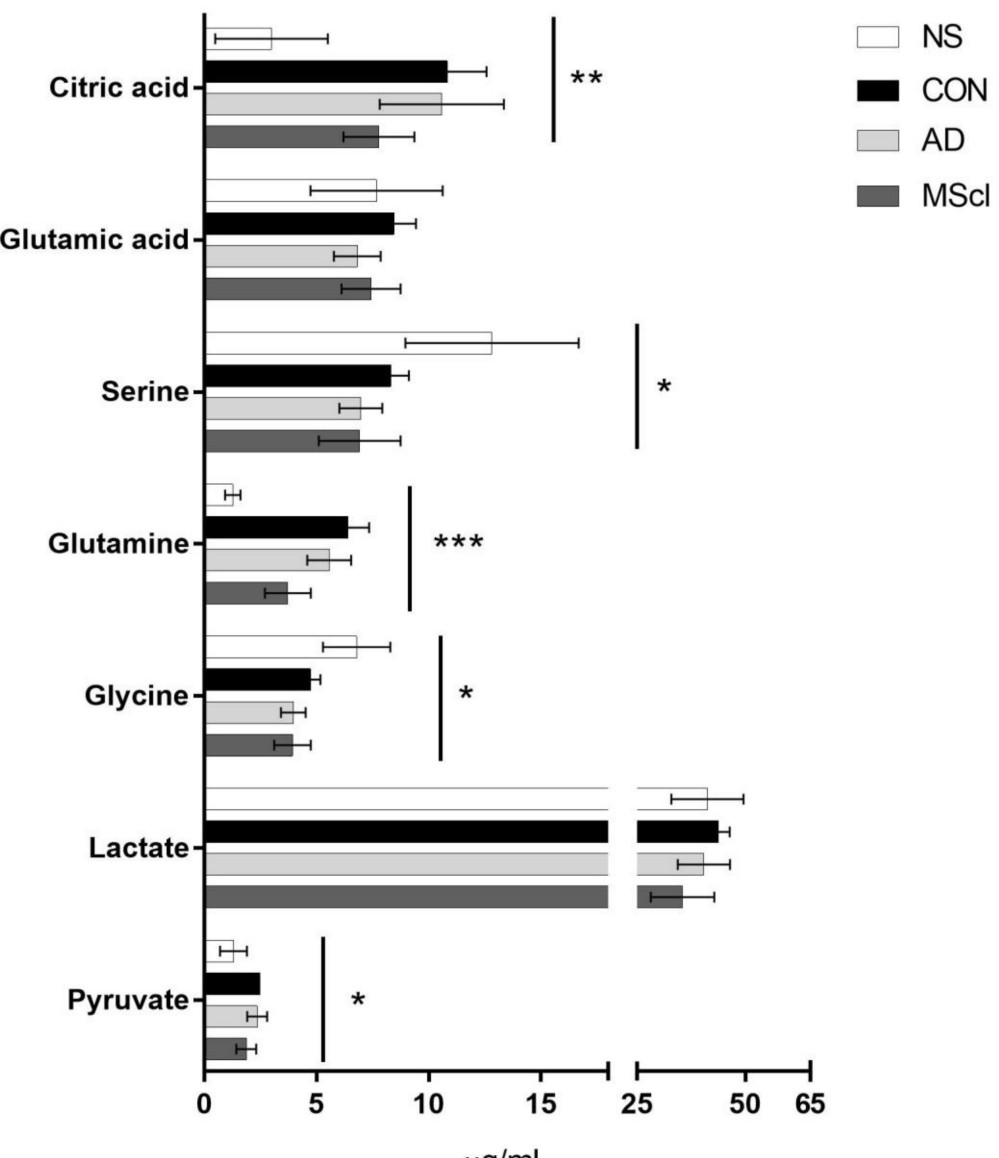

**Figure 2.** Concentrations of unlabeled analytes detected and quantified in the cell extract (intracellular amounts) of $8 \times 10^5$ monocytes/well, incubated for 5 h in four different conditions: non-stimulation (NS, 1), treatment with CSF from healthy donors (CON, 2), patients with AD (3), or MScl (4). The figure shows the *p*-values of all four groups obtained with one-way ANOVA for data normally distributed or Kruskal–Wallis test for data non-normally distributed (* $p \leq 0.05$; ** $p \leq 0.01$; *** $p \leq 0.001$). Tukey's test or Dunn's test (non-parametric) were performed as post hoc tests.

Similar to the unlabeled endogenous glucose metabolites, the majority of 1,2-$^{13}C_2$ glucose-derived compounds were 2,3-$^{13}C_2$ pyruvate and 1,2-$^{13}C_2$ lactate. A small amount of $^{13}C$-glutamine was quantified in CSF-treated monocytes (>LOQ), while its concentration was below LOQ in untreated monocytes (Figure 3).

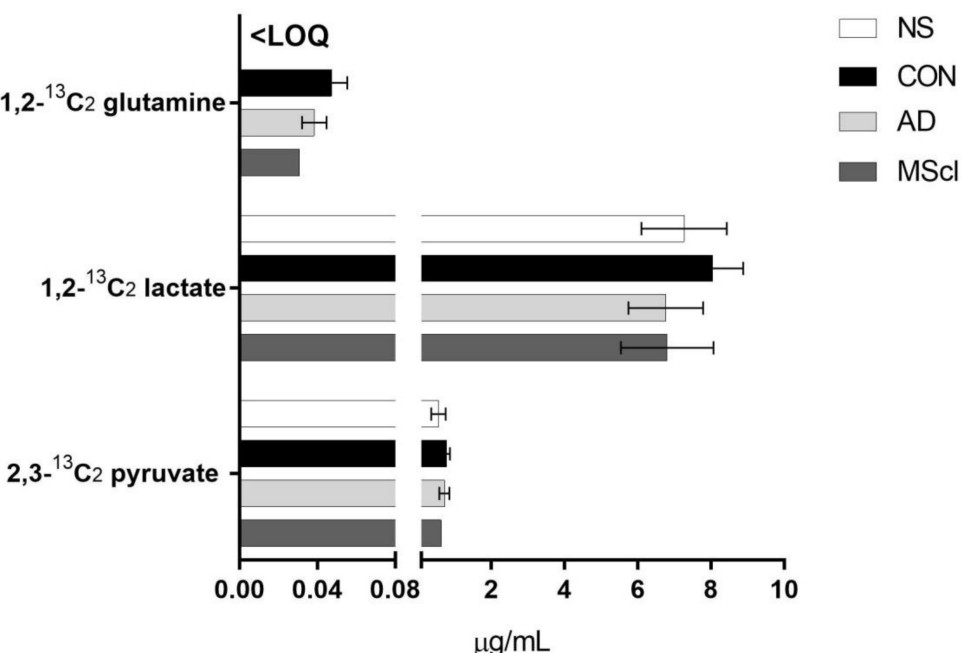

**Figure 3.** Incorporation of [13]C starting from 1,2-[13]C$_2$ glucose into glucose-metabolites in the analytes quantified from the cell extract after 5 h of incubation of $8 \times 10^5$ monocytes/well. In this case, no significant differences were found between the treatment groups (one-way ANOVA for data normally distributed or Kruskal–Wallis test for data non-normally distributed), with an exception made for labeled glutamine that was not quantifiable in NS. <LOQ: the concentration of the analyte was less than the limit of quantitation.

### 3.2. Differential Concentration of Secreted Glucose-Derived Metabolites

Next, the same analysis was performed with the cultured medium to quantify glucose-derived metabolites that were secreted into the culture medium. This information gave an overview of the overall conversion of glucose as well as [13]C-glucose into its metabolites under the applied experimental conditions. The concentrations of unlabeled and [13]C-labeled metabolites are shown in Figures 4 and 5, respectively.

Similar to the results obtained from cellular fraction (i.e., intracellular metabolites), significant differences in the level of citric acid ($p = 3.2 \times 10^{-6}$), glutamine ($p = 9.0 \times 10^{-7}$), pyruvate ($p = 6.9 \times 10^{-5}$), serine ($p = 1.5 \times 10^{-10}$), and glycine ($p = 1.3 \times 10^{-6}$) were detected between the NS and the CSF-treatment groups (Table 1). Glutamic acid ($p = 4.6 \times 10^{-5}$) shows significant variations in concentrations only between NS and CSF of patients (AD and MScl). There are also some significant differences between CSF-treated groups. Lactate ($p = 0.0125$), serine, and glutamic acid displayed a significant variation between CON and AD, whereas lactate, glutamine, and glutamic acid differed between CON and MScl and glutamine, and serine between AD and MScl (Figure 4).

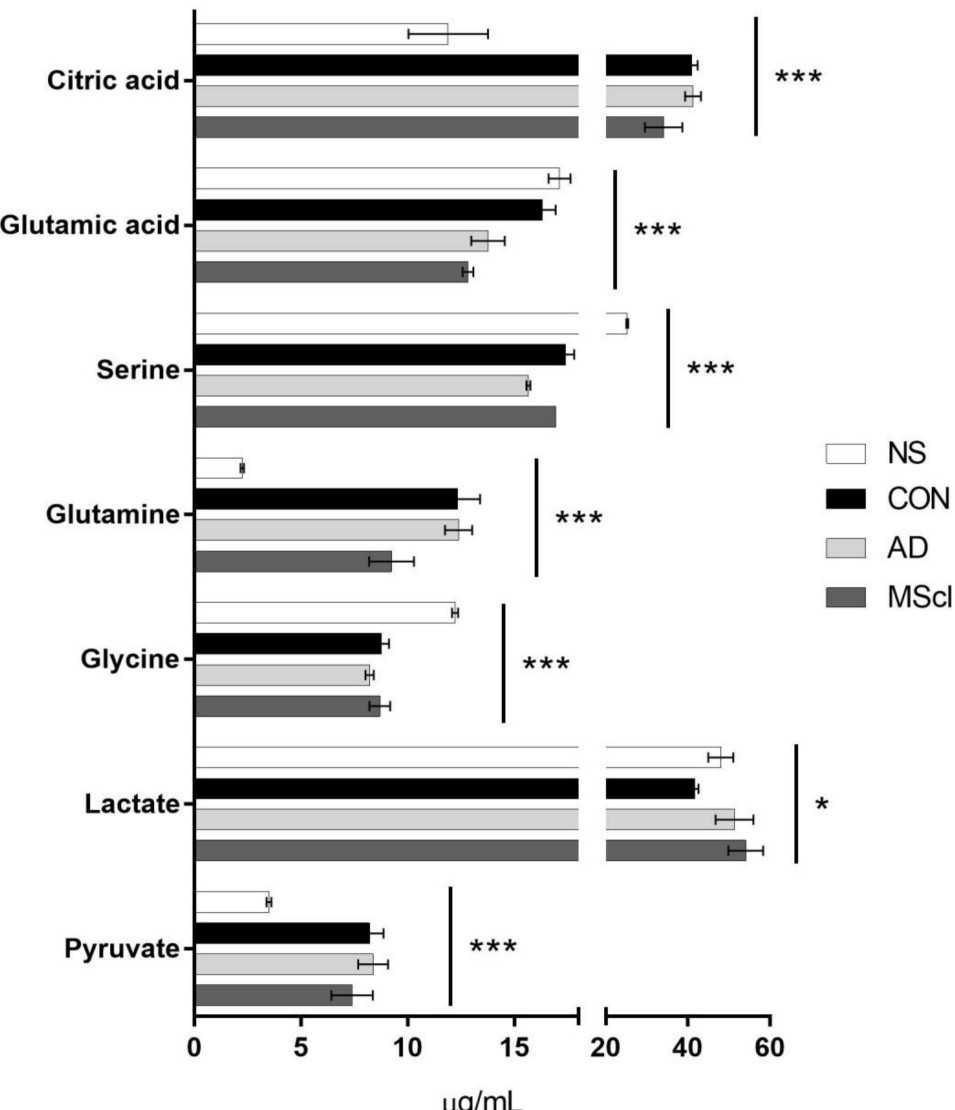

**Figure 4.** Concentrations of analytes detected and quantified in the culture medium (extracellular amounts) of $8 \times 10^5$ monocytes/well. The figure shows the *p*-values for all four groups obtained with one-way ANOVA for data normally distributed or Kruskal–Wallis test for data non-normally distributed (* $p \leq 0.05$; *** $p \leq 0.001$). Tukey's test or Dunn's test (non-parametric) were performed as post hoc tests.

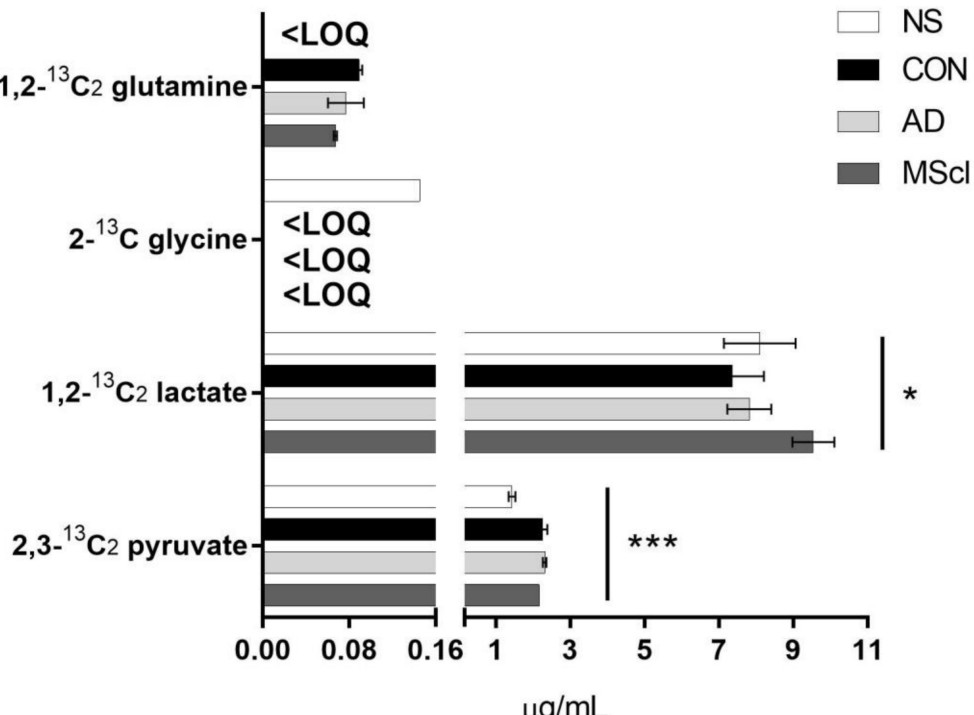

**Figure 5.** Incorporation of $^{13}C$ starting from 1,2-$^{13}C_2$ glucose into glucose-metabolites in the analytes quantified in the cell incubation medium after 5 h of incubation of $8 \times 10^5$ monocytes/well. In the figures are shown the *p*-values obtained with one-way ANOVA for data normally distributed or Kruskal–Wallis test for data non-normally distributed (* $p \leq 0.05$; *** $p \leq 0.001$). Tukey's test or Dunn's test (non-parametric) were performed as post hoc tests. <LOQ: the concentrations of the analytes were less than the limit of quantitation.

Determination of the incorporation of $^{13}C$ from 1,2-$^{13}C_2$ glucose revealed the majority was converted to pyruvate (2,3-$^{13}C_2$ pyruvate) and lactate (1,2-$^{13}C_2$ lactate), and only in small amounts to glutamine (1,2-$^{13}C_2$ glutamine) and glycine (2-$^{13}C$ glycine), as shown in Figure 5. 2,3-$^{13}C_2$ serine and 1,2-$^{13}C_2$ citric acid were detected in all samples, but the concentrations were lower than LOQ, and thus they are not shown in Figure 5. Furthermore, 1,2-$^{13}C_2$ glutamine could only be quantified in the monocytes treated with CSF and 2-$^{13}C$ glycine only in the NS group. Differential abundances of 2,3-$^{13}C_2$ pyruvate and 1,2-$^{13}C_2$ lactate were significantly increased in CSF-treated monocytes (*p*-values of $7.9 \times 10^{-5}$ and 0.0318, respectively).

To summarize the significant differences between all groups, Table 1 reports the *p*-values obtained with the one-way ANOVA (for normally distributed) or the Kruskal–Wallis test (non-parametric) and the results of the Tukey's or Dunn's tests to establish the differences between each group.

**Table 1.** Significant differences between groups. The *p*-values were obtained with the one-way ANOVA test for normally distributed data or with the Kruskal–Wallis test for non-normally distributed data. The asterisks show which group was significantly different from the other. Results were obtained with post hoc tests (Tukey's test for normally distributed and Dunn's test for non-parametric).

| Analyte | Cell Lysate | | | | | | | Incubation Medium | | | | | | |
|---|---|---|---|---|---|---|---|---|---|---|---|---|---|---|
| | *p*-value | NS vs. CON | NS vs. AD | NS vs. MScl | CON vs. AD | CON vs. MScl | AD vs. MScl | *p*-value | NS vs. CON | NS vs. AD | NS vs. MScl | CON vs. AD | CON vs. MScl | AD vs. MScl |
| Pyruvate | 0.0387 | # | | | | | | $6.9 \times 10^{-5}$ | # | # | # | | | |
| 2,3-$^{13}$C$_2$ pyruvate | 0.3611 | | | | | | | $7.9 \times 10^{-5}$ | # | # | # | | | |
| Lactate | 0.4955 | | | | | | | 0.0125 | | | | | # | # |
| 1,2-$^{13}$C$_2$ lactate | 0.3963 | | | | | | | 0.0318 | | | | | # | |
| Glycine | 0.0166 | | * | * | | | | $1.3 \times 10^{-6}$ | * | * | * | | | |
| 1,2-$^{13}$C$_2$ glutamine | 0.2199 | | | | | | | 0.0665 | | | | | | |
| Glutamine | 0.0004 | # | # | # | | * | | $9.0 \times 10^{-7}$ | # | # | # | | * | * |
| Serine | 0.0356 | | * | * | | | | $1.5 \times 10^{-10}$ | * | * | * | * | | # |
| Glutamic acid | 0.7333 | | | | | | | $4.6 \times 10^{-5}$ | | * | * | * | * | |
| Citric acid | 0.0073 | # | # | | | | | $3.2 \times 10^{-6}$ | # | # | # | | | |

# Significantly different with the post hoc tests. The first term of comparison is lower than the second. * Significantly different with the post hoc tests. The first term of comparison is higher than the second.

### 3.3. Endogenous Glucose-Derived Metabolites in CSF

We took into consideration that there are some naturally abundant metabolites in CSF, e.g., lactate, pyruvate, and glutamine. These compounds were also detected and quantified. In Figure 6, we compare these CSF-metabolites to the metabolites detected in monocytes and the medium after incubation. Here, we considered only glutamine, citric acid, lactate, and pyruvate, the metabolites that were significantly different between the groups as shown above. The total amounts of intracellular and secreted metabolites are much higher than those naturally present in CSF, confirming the metabolic activity of monocytes and the further production of these compounds (i.e., both unlabeled and $^{13}$C-labeled) during the incubation. The concentration of glucose in CSF of healthy individuals or patients with AD or MScl is reported to be similar [48–51]. Moreover, the total concentrations of glucose (labeled and not labeled) during the incubation are quite similar between groups (i.e., NS, CON, AD, and MScl).

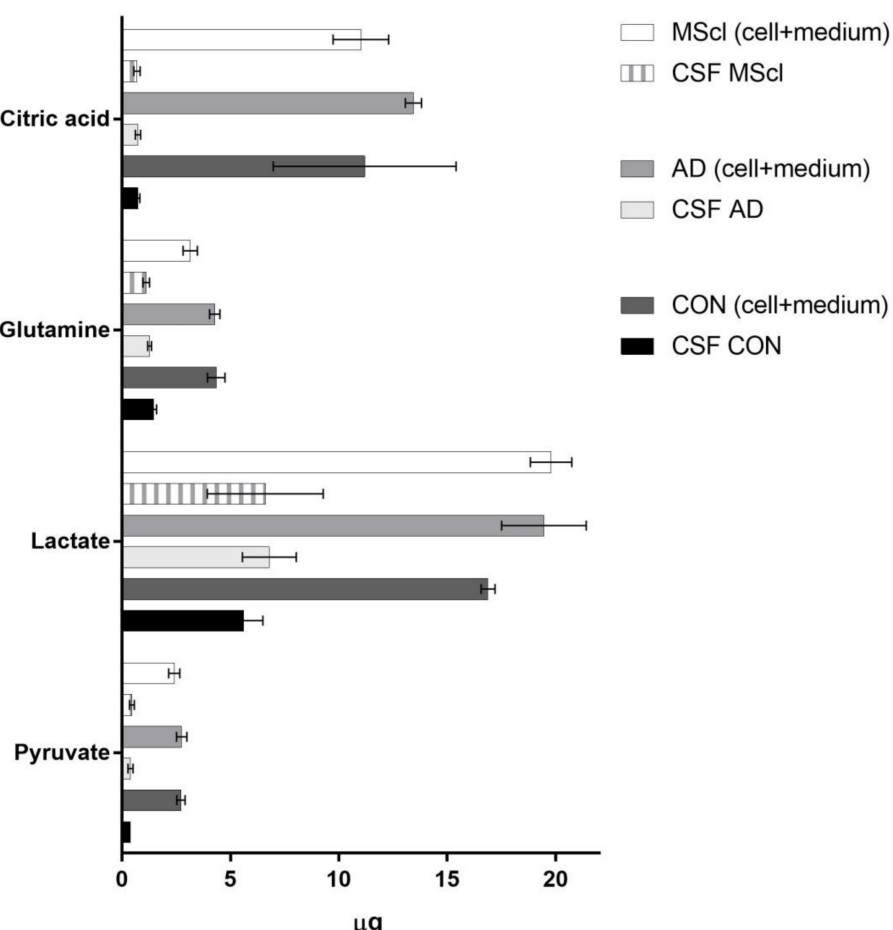

**Figure 6.** Comparison of the amounts of the endogenous metabolites detected in CSF with the sum of intra- and extracellular. CSF MScl: μg of analytes quantified in the CSF of MScl patients; MScl (cell + medium): sum of the μg of analytes quantified in the cell lysate and medium of monocytes incubated with MScl-CSF; CSF AD: μg of analytes quantified in the CSF of AD patients; AD (cell + medium): sum of the μg of analytes quantified in the cell lysate and medium of monocytes incubated with AD-CSF; CSF CON: μg of analytes quantified in the CSF of healthy donors. CON (cell + medium): sum of the μg of analytes quantified in the cell lysate and medium of monocytes incubated with CON-CSF. The values are the mean of three biological replicates ± SD.

## 4. Discussion

The incorporation of $^{13}$C from labeled glucose into its metabolites allows insights into the metabolic pathway that was preferentially taken during in vitro stimulation. Further to small amounts of the remaining endogenous glucose present in the monocytes or the CSF, the only source of glucose was 1,2-$^{13}$C$_2$-glucose added to the glucose-free medium. Primary human monocytes isolated from the peripheral blood of healthy individuals were used in this study. The main limitation of our study was the number of monocytes obtained from a limited volume of the peripheral blood, which in turn resulted in low amounts of detected metabolites. However, the feasibility to apply this method to study changes in cellular metabolism (i.e., glucose) in a small sample of human primary cells was demonstrated. Although not all glucose-derived metabolites could be detected due to the limitation mentioned above, an increased level of both unlabeled and/or labeled citric acid, glutamine, and pyruvate in CSF-treated monocytes and in the culture medium was consistently detected, whereas serine and glycine were consistently reduced after treatment with CSF. Our findings suggest the rewiring of the glucose metabolic pathway in monocytes after treatment with CSF, possibly due to an increased cellular activation. Commonly, activation of human innate immunity requires alteration of cellular metabolic pathways largely to favor glucose metabolism [52–56].

The significant increase in the citric acid concentration, once monocytes came into contact with CSF no matter from which group, follows the increase in pyruvate mentioned above, suggesting a more active tricarboxylic acid (TCA) cycle as well. Due to the limited amount of labeled pyruvate, and the preferential pathway towards the formation of lactate, it is not a surprise that other labeled compounds of the TCA cycle were not quantifiable. Moreover, rate-limiting in the TCA cycle is the citrate synthase catalyzed step [44]. As labeled citric acid could be detected but not quantified (>LOD but <LOQ), the analysis of other labeled metabolites of the TCA cycle is difficult. Furthermore, it must be considered that the TCA cycle will undergo multiple cycles within 5 h of incubation. Thus, it is reasonable to expect a fast interconversion and consumption of the other metabolic intermediates and a production of a variety of labeled metabolites. Since the aim of this study was to gain an overview of the metabolic trends, only the analysis of the first labeled product is considered. The results of the 1,2-$^{13}$C$_2$ citric acid are displayed in neither Figure 3 nor Figure 4 since they were <LOQ. Comparing the chromatographic peak areas, though, a similar trend as for the unlabeled citric acid was observed.

As highlighted in the introduction, monocytes undergo phenotypic changes and contribute to the progress of inflammation. As this requires more energy, metabolic reprogramming is observed. The results shown in this study corroborate this hypothesis. In line with another study by Ren et al. [57], monocytes and macrophages increase their glycolysis during their phenotypic conversion, e.g., from homeostatic to inflammatory state [58–60]. The role of citric acid in CSF, though, is multifaceted. It is a good chelating agent (therefore, a challenging analyte for LC analysis) and many studies correlated its concentrations in CSF to calcium, magnesium, or zinc cations [61,62]. Infantino et al. showed that citrate and the mitochondrial citrate carrier play a significant role in inflammation [63,64]. However, at present, it is not possible to discriminate between potential reasons for increased concentrations of citric acid in CSF-treated groups. Even if it is safe to hypothesize that there is increased glycolytic activity to support cell activation and phenotypic changes, it is not excluded that its role may be more complex and correlated to the new compartment.

Increased glucose-derived glutamine (both unlabeled and $^{13}$C-labeled) in monocytes treated with CSF confirms an enhancement of the TCA cycle in CSF-treated monocytes. However, this amino acid is also present in CSF. Thus, it serves as an extra source of glutamine that is not present in the NS group, which was incubated in a glutamine-free medium. However, there is a clear increase in glutamine amounts in cell extract and medium compared with CSF only (Figure 6). Moreover, the MScl group presents significant differences also with CON in the cell extract (Figure 2), and with CON and AD in the medium (Figure 4), suggesting different activation phenotypes in monocytes from patients

with neuroinflammatory conditions such as MScl to those with neurodegeneration such as AD.

As shown above, the level of glutamic acid was significantly decreased in monocytes treated with CSF from patients with AD and MScl, whereas treatment with CON-CSF resulted in a similar level of glutamic acid. This supports the finding that, in the presence of CSF, monocytes show more likely reactive phenotypes, indicated by an increased glucose conversion via pyruvate and citric acid towards glutamine (possibly through the TCA cycle), especially for healthy individuals and, to a slightly lesser extent, for patients with lower neuroinflammatory conditions (AD). This also implies the need for energy (e.g., via ATP production) during phenotypic alteration and a differentiation in the metabolism rewiring between AD and MScl neuroinflammatory conditions. Figure 7 summarizes these changes; in red are highlighted the metabolites that present a decrease in concentration after the stimulation with CSF, and in blue are those that show an increase.

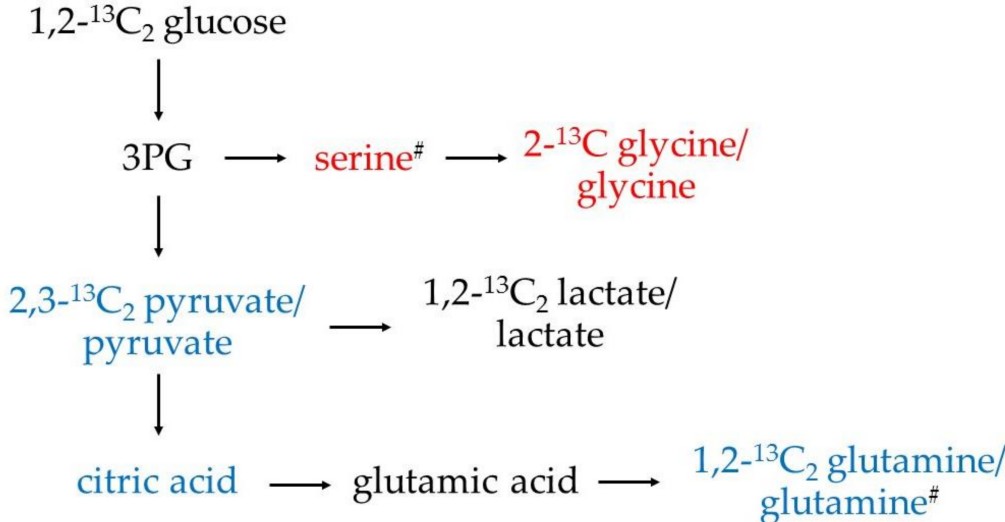

**Figure 7.** Alteration of metabolic pathways of the cell. In red are highlighted the metabolites that presented a significant decrease in the concentration when treated with CSF, and in blue, those that presented a significant increase. $^{13}C$ citric acid and $^{13}C$ glutamic acid were detectable but not quantifiable because of their fast interconversion towards $^{13}C$ glutamine, and therefore, are not displayed. The metabolites highlighted with "#" are significantly different between diseases (ADvsMScl), suggesting different activation phenotypes between AD and MScl.

Furthermore, glutamate is neurotoxic at certain concentrations in CSF, although it is essential for the homeostasis of the body. Conversion of glutamate to glutamine is one of the mechanisms to eliminate the excess of glutamate to maintain system homeostasis. Of note, MScl patients usually show elevated levels of glutamate in the brain, leading to excitotoxicity [65–67]. However, as shown in this study, this is not the case for monocytes in the CSF environment. Moreover, low levels of glutamate in the parietal and cingulate regions and right hippocampus were correlated to loss of memory and cognitive impairment [68]. Moreover, glutamine/glutamate levels are connected to nitric oxide production. They may also contribute to the formation of glutathione and play a role in antioxidant defenses [69].

The CSF-stimulated groups exhibited lower concentrations of glycine and serine in comparison with NS. The pathway towards the production of these amino acids passes through 3-phosphoglyceric acid (3PG). Thus, monocytes, when stimulated with CSF, diverge the metabolism of 3PG towards the production of pyruvate instead of glycine and serine. Under LPS-stimulation, glycine shows modulatory effects on cytokine production of monocytes, that is, a reduction in TNF-α and IL-1β expression [70]. These results show that, under the applied experimental conditions (with CSF treatment), monocytes adapted their reactive phenotypes by increasing glycolysis via pyruvate conversion to glutamine

and reducing other metabolic pathways such as serine and glycine conversion. The levels of serine also differ between AD and MScl groups, suggesting again different phenotypic alterations between the considered diseases.

## 5. Conclusions

In summary, we report a significant alteration in the metabolism of monocytes when incubated with CSF. Overall, the metabolism showed a preference towards the TCA cycle and production of glutamine for the cells incubated with CSF, especially those from patients with AD and MScl. Other metabolic pathways such as serine and glycine production were down-regulated. These changes in glycolysis imply the alteration of monocyte phenotypes towards a reactive state. However, our study has some limitations, including: (1) it presents a limited number of biological replicates due to the rarity of the specimen CSF considered; (2) it uses an in vitro experimental model that may not completely explain the in vivo situation, and; (3) due to low cell numbers, some concentrations of the metabolites were lower than the limit of detection and/or limit of quantitation. Therefore, optimization and further development of the analytical methods are required, including the use of labeled glutamine for the parallel characterization of TCA cycle metabolites [71,72] and the use of a completely bioinert LC-MS/MS system.

Understanding how blood monocytes respond to the local environment (e.g., changing compartment from blood to CSF) at both phenotypic and functional/metabolic states will provide further insights on how to better regulate these cells, especially in neurological diseases, in which these cells play an important role [73,74].

**Supplementary Materials:** The following supporting information can be downloaded at: https://www.mdpi.com/article/10.3390/cimb45010051/s1, Table S1: Concentrations of the targeted analytes in cell lysate and culture medium. The reported concentrations are mean ($\pm$ SD) values of three biological replicates.

**Author Contributions:** Conceptualization, G.G., M.K.P. and C.B.; methodology, G.G.; formal analysis, G.G.; investigation, G.G.; resources, C.B. and M.K.P.; writing—original draft preparation, G.G.; writing—review and editing, C.B. and M.K.P.; visualization, G.G.; supervision, M.K.P. and C.B.; project administration, C.B., M.K.P. and G.G.; funding acquisition, C.B., M.K.P. and G.G.; recruitment of the patients and access to biomaterials (CSF), J.P., K.R. and C.O. All authors have read and agreed to the published version of the manuscript.

**Funding:** G.G. and C.B. were funded by the German Research Foundation (DFG)—Project-ID 259373024—CRC/TRR 167 (B05). G.G. was additionally funded by the State of Berlin, Germany, with the Elsa Neumann Ph.D. scholarship (Antrags Nr.: 069032). Article processing charges (APC) are covered by the Open Access Publication Initiative of Freie Universität Berlin. K.R. received research support from Novartis Pharma, Merck Serono, German Ministry of Education and Research, European Union (821283-2), Stiftung Charité (BIH Clinical Fellow Program), and Arthur Arnstein Foundation, and received speaker honoraria from Bayer and travel grants from Guthy Jackson Charitable Foundation.

**Institutional Review Board Statement:** The study was approved by the Ethics Committee of Charité—Universitätsmedizin Berlin (Ethikkommission der Charité—Universitätsmedizin Berlin;), registration number EA1/187/17.

**Informed Consent Statement:** Informed consent was obtained from all subjects involved in the study.

**Data Availability Statement:** Not applicable.

**Acknowledgments:** We would like to acknowledge the assistance of the Core Facility BioSupraMol supported by the DFG. The authors thank Bernhard Wüst, Agilent Technologies Inc., for his helpful support in mass spectrometry, and Adeline Dehlinger and Christian Böttcher for the sample collection of CSF and PBMCs. We acknowledge support by the OpenAccess Publication Fund of Freie Universität Berlin.

**Conflicts of Interest:** The authors declare no conflict of interest. The funders had no role in the design of the study; in the collection, analyses, or interpretation of data; in the writing of the manuscript, or in the decision to publish the results.

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
