# Peer review of "1,2-13C2-Glucose Tracing Approach to Assess Metabolic Alterations of Human Monocytes under Neuroinflammatory Conditions"

_cimb, doi:10.3390/cimb45010051_

Round 1

Reviewer 1 Report

In the current manuscript authors investigated the effect of after exposure to CSF of healthy individuals or patients with AD or MScl on the metabolic pathway on blood monocytes, to get more information about involvement of neuroinflammation in the pathogenesis of neurological disorders. They found higher pIn the current manuscript authors investigated the effect of after exposure to CSF of healthy individuals or patients with AD or MScl on the metabolic pathway on blood monocytes, to get more information about involvement of neuroinflammation in the pathogenesis of neurological disorders. They found higher production of citric acid and glutamine, suggesting a more active glycolysis and tricarboxylic acid (TCA) cycle, and reduced production of glycine and serine after exposure to the CSF from patients and healthy donors. It is an interesting study, however, there are several overlapping with the previous published study from the same group and it is not clear what is the novelty of this study compared with the previous one.

Author Response

Point-by-point answer to Reviewer 1. 

Thank you very much for your comments, please see the attachment for the response.

Reviewer 2 Report

The authors have analyzed neuro-inflammation and the metabolic status of monocytes in CON and AD individuals in comparison to MScI.

The study is innovative and very elegant.

Minor comments:

Please proof read for grammatical error.

Author Response

Reviewer 2

We want to thank you for your work, suggestions for the paper, and nice comments!

We checked the manuscript again for grammatical errors.

Round 2

Reviewer 1 Report

The information that authors provided  and added into the manuscript, made a clear description of study novelty.